# Growth Factor-Primed WJ-MSC Secretome Enhances Fibroblast Expansion In Vitro

**DOI:** 10.3390/biomedicines13122863

**Published:** 2025-11-24

**Authors:** Katia Jarquín-Yáñez, Diana Michel Aguilar-Sandoval, Gabriela Piñón-Zárate, Miguel Angel Herrera-Enríquez, Ivan Daniel Avila-Campos, Beatriz Hernandez-Tellez, Dayana Mayte Dominguez-Diaz, Blanca Esther Blancas-Luciano, Andrés Eliú Castell-Rodríguez

**Affiliations:** 1Laboratory of Immunotherapy and Tissue Engineering, Department of Cellular and Tissue Biology, Faculty of Medicine, National Autonomous University of Mexico (UNAM), Av. Universidad 3000, Copilco Universidad, Coyoacán, Ciudad de México 04510, Mexico; jy.katy@facmed.unam.mx (K.J.-Y.); acuaporina08@gmail.com (D.M.A.-S.); gabrielapinon@unam.mx (G.P.-Z.); mikeh@unam.mx (M.A.H.-E.); ivan.avicamp@gmail.com (I.D.A.-C.); bhdezt@unam.mx (B.H.-T.); daytediaz@ciencias.unam.mx (D.M.D.-D.); 2Department of Microbiology and Parasitology, Faculty of Medicine, National Autonomous University of Mexico (UNAM), Ciudad de México 04510, Mexico; bblancas@facmed.unam.mx

**Keywords:** wharton’s jelly cells (WJ-MSC), epidermal growth factor (EGF), fibroblast growth factor (FGF), secretome

## Abstract

**Background:** In regenerative medicine, there is interest in using acellular therapy based on the secretome of mesenchymal stem cells (MSC) to promote wound healing. Wharton’s jelly cells (WJ-MSCs) are a readily available source. Their secretion has been optimized when stimulated with FGF and EGF to induce proliferation and prevent senescence. Therefore, evaluating the effect on proliferation and wound closure of human fibroblasts in vitro with different concentrations of the secretome of WJ-MSCs stimulated with growth factors is necessary to identify the most efficient work concentration. **Methods:** The secretome of human WJ-MSC was collected from passage 1 to passage 2 stimulated with FGF and EGF (W FGF/EGF) and the unstimulated secretome (WO FGF/EGF). The immunophenotype of WJ-MSCs after stimulation was evaluated by flow cytometry for the markers: CD105+, CD73+, CD90+, HLA-ABC+, CD44+, HLA-DR−, CD34−, CD11b−, CD19−, and CD45−. The presence of 14 growth factors in the secretome was evaluated using LEGENDplex through flow cytometry. Fibroblasts were cultured, and their culture medium was supplemented with two different concentrations: one of 1.25 mg/mL and another of 6.25 mg/mL of both stimulated and unstimulated secretome. Proliferation, cellular metabolism, and wound closure were evaluated in vitro. **Results:** The immunophenotype of WJ-MSCs after stimulation remained unchanged, and the production of growth-assessed factors was increased in stimulated WJ-MSCs. The optimal concentration that induced proliferation and wound closure in vitro was 1.25 mg/mL of stimulated WJ-MSC secretome. **Conclusions:** This study demonstrates that stimulation of WJ-MSCs with FGF and EGF enhances the secretion of growth factors, and that a concentration of 1.25 mg/mL of their secretome promotes optimal fibroblast proliferation and wound closure in vitro. These findings support the potential of optimized WJ-MSC secretome as a promising acellular strategy for regenerative medicine.

## 1. Introduction

Wharton’s jelly mesenchymal stem cells (WJ-MSCs) from umbilical cord tissue hold considerable promise for regenerative medicine, owing to their broad differentiation potential and potent immunomodulatory activity [1,2,3,4]. WJ-MSCs display a fibroblast-like morphology, great adherence to plastic, and robust expression of characteristic surface markers, such as CD90, CD73 and CD105, which correspond to the expression of the NT5E, Thy1, and ENG genes, respectively, and CD44, with low levels of MHC class I (HLA A, B, C) and no expression of the hematopoietic markers CD34, CD45, CD19, CD11b, or MHC class II. In addition, they could express pluripotent embryonic stem cell markers such as Nanog, Oct4 and Sox2 at low levels. It reinforces their regenerative potential and their role in tissue differentiation and repair processes. The WJ-MSCs can differentiate into osteogenic, adipogenic, and chondrogenic lineages [5,6,7,8].

The clinical advantages of WJ-MSCs are well documented, with both autologous and allogeneic transplants demonstrating efficacy across diverse conditions [9,10]. However, a significant challenge in their therapeutic application is the requirement for repeated administrations to achieve complete tissue regeneration. In models of spinal cord injury, osteoarthritis, and diabetes, multiple doses have yielded markedly improved functional recovery [11,12,13]. This need for repeated dosing poses practical limitations, including the requirement for continuous in vitro expansion to obtain enough cell numbers. In addition, long-lasting cell cultures can compromise therapeutic efficacy by promoting genetic instability or accelerating cellular senescence, reducing differentiation potential after successive passages [14,15,16].

One of the primary mechanisms by which WJ-MSCs promote tissue regeneration is through the release of bioactive factors—collectively termed the secretome—which has recently emerged as a potential alternative to whole-cell therapies in regenerative medicine. The secretome comprises soluble growth factors, signaling molecules, and extracellular microvesicles, all of which are released into the surrounding medium and can be harnessed in cell-free approaches for regenerative, immunomodulatory, neuroprotective, and proangiogenic applications, among others [17,18,19,20].

Significantly, the composition of the secretome varies according to cell type, culture conditions, and donor age. It can also be modulated in vitro, providing an opportunity to tailor its composition for different therapeutic purposes or to combine it with secretomes from other cell types, thereby enriching the levels of growth factors for specific clinical applications. Building on this concept, recent research has focused on the mechanical and chemical cues that regulate cytokine and growth factor secretion by WJ-MSCs and other MSCs, given their modulatory and protective roles in tissue homeostasis [21,22,23,24,25]. In this context, “cell preconditioning” strategies—such as hypoxia or the administration of drugs, hormones, and growth factors—are being explored to optimize secretome composition by reshaping the culture environment and influencing the profile of secreted factors [26,27].

Among the different growth factors, Epidermal Growth Factor (EGF) is the most critical mediator of epithelial repair, by driving cell migration, differentiation, and tissue remodeling. In addition, it accelerates wound healing by maintaining homeostasis in damaged tissues [28], whereas its absence delays repair and favors hypertrophic scarring [29].

Alongside EGF, Fibroblast Growth Factor (FGF) is essential for mesenchymal cell proliferation and secretory activity. FGF also preserves the multipotent phenotype of mesenchymal stem cells in vitro and in vivo, enhancing their stability and differentiation potential [30,31]. Both FGF and EGF are widely applied to expand MSCs, as they sustain proliferation, prevent senescence, and promote secretion of regenerative and immunomodulatory factors. Therefore, this study aimed to assess the change in concentration of certain growth factors in the secretome of WJ-MSCs stimulated with FGF and EGF, as well as their activity on human fibroblast proliferation in vitro.

## 2. Materials and Methods

### 2.1. Sample Collection and Cord Processing

Umbilical cord samples (10–15 cm) were obtained with informed consent from cesarean deliveries at the Magdalena Contreras Maternal and Child Hospital and approved by the Ethics and Research Committees of the Faculty of Medicine, UNAM (protocol 501-010-18-24 and FM/DI/112/2022). Samples were aseptically collected, placed in DMEM-F12 medium (Dulbecco’s Modified Eagle Medium: Nutrient Mixture F12, Gibco, Waltham, MA, USA), supplemented with 2% antibiotic (penicillin, streptomycin and amphotericin B, Gibco, Waltham, MA, USA), stored at 4 °C, and then transported to the laboratory for processing.

The explant method was used to isolate mesenchymal stem cells from the Wharton’s jelly of the umbilical cord [32]. Briefly, umbilical cords were washed three times with Hank’s saline solution containing 1% antibiotic (penicillin, streptomycin and amphotericin B, Gibco, Waltham, MA, USA), and vessels were removed. The cords were cut into 3–5 mm fragments, cultured in Petri dishes with DMEM-F12 medium (10% FBS, 1% antibiotic (penicillin, streptomycin and amphotericin B, Gibco, Waltham, MA, USA) at 37 °C with 5% CO_2_, and the medium was replaced every third day. After 10 days, WJ-MSCs migrated and adhered to the plastic, and tissue fragments were discarded. At 90% confluence (P0), cells were detached with 0.05% trypsin–0.02% EDTA (Trypsin-EDTA, Gibco, Waltham, MA, USA), expanded to P1 in 75 cm^2^ flasks, and stored for flow cytometry phenotyping.

### 2.2. Stimulation and Secretome Collection from WJ-MSC Cultures

WJ-MSC (P1) cultures at 80% confluence were stimulated with 10 ng/mL of EGF and 40 ng/mL of FGF for 48 h (EGF and FGF Human, Sigma-Aldrich, St. Louis, MO, USA). According to Park et al. (2009) [33], these concentrations were effective in inducing stem cell proliferation. The growth factors were prepared in plain medium DMEM-F12 [33], after which WJ-MSC were maintained in plain medium for 48h until secretome collection. 10 mL of supernatant of WJ-MSC yielded 0.3235 g of lyophilized powder. The secretome was frozen at −70 °C and lyophilized at 0.02 hPa for 24 h (Cylindrical Coolsafe Freeze Dryer, Labogene, Lillerød, Denmark). Cells were detached with trypsin–EDTA; 5 × 10^6^ cells were stored for flow cytometry, and 1 × 10^6^ cells were expanded to P2, repeating the procedure to obtain more secretome.

### 2.3. Flow Cytometry Assay

Unstimulated and growth factor–stimulated P1 and P2 cells were incubated with fluorochrome-conjugated (fluorescein isothiocyanate-FITC and phycoerythrin-PE) stained with conjugated antibodies against CD105 (no. cat. 323203), CD90 (no. cat. 328109), CD73 (no. cat. 328109), CD44 (no. cat. 397503), HLA-DR (no. cat. 361605), HLA-ABC (no. cat. 397517), CD34 (no. cat. 378603), CD11b (no. cat. 982606), CD45 (no. cat. 384407) and CD19 (no. cat. 302207) (BioLegend, San Diego, CA, USA). Cells were detached with trypsin–EDTA, incubated with antibodies at a 1:300 dilution for 30 min at room temperature in the dark, washed, and then fixed with PBS/1% paraformaldehyde. The cells were analyzed by flow cytometry (Attune NxT, BD, Franklin Lakes, NJ, USA), as performed at LabNalCit UNAM. Data were processed using FlowJo v8.8.7 (BD, Franklin Lakes, NJ, USA).

### 2.4. Multiplex Analysis of WJ-MSC Secretome

To characterize the secretome of WJ-MSCs, the collected and lyophilized supernatant from P1 and P2 cultures was concentrated to a minimal concentration of 100 mg/mL for LEGENDplex detection. The concentration quantification was performed using the LEGENDplex Human Growth Factor Multiplex Assay according to the manufacturer’s instructions (LEGENDplex, BioLegend, San Diego, CA, USA). Fourteen factors—Ang-2, EGF, EPO, FGF, G-CSF, GM-CSF, HGF, M-CSF, PDGF-AA, PDGF-BB, SCF, TGF-α, TGF-β, and VEGF—were simultaneously measured in the secretome of both factor-stimulated and unstimulated WJ-MSCs. Beads were incubated with each group in the experiment and analyzed on an Attune NxT flow cytometer (BD, Franklin Lakes, NJ, USA). Data were processed using BioLegend’s LEGENDplex v8.0 software, and results were reported as pg/mL.

### 2.5. Isolation and Primary Culture of Human Dermal Fibroblasts

Human fibroblasts were obtained from split-thickness skin biopsies collected from the inner forearm using a 5-mm diameter punch under aseptic conditions. All tissues were voluntarily donated by healthy male individuals with an age range between 25 and 40 years old, who provided informed consent approved by the institutional research and ethics committees (FM/DI/028/2024). Skin biopsies were washed three times by immersion in sterile HBSS containing 1% antibiotic (penicillin, streptomycin and amphotericin B, Gibco, Waltham, MA, USA) at 4 °C for 10 min, and the dermal and epidermal layers were enzymatically separated by incubation in 1.2 U/mL dispase at 37 °C for 60 min. Dermal sheets were then fragmented into 1–4 mm^2^ specimens, seeded in culture dishes, and incubated at 37 °C with 5% CO_2_ in DMEM supplemented with 10% FBS and 1% antibiotic (penicillin, streptomycin and amphotericin B, Gibco, Waltham, MA, USA). The medium was replaced every third day, and the fragments were left undisturbed to allow fibroblasts to migrate and adhere to the plastic. When passage 0 (P0) cells reached 90% confluence, they were detached using 0.05% trypsin –0.02% EDTA and expanded in new plates.

### 2.6. Effect of WJ-MSC Secretome on Cytotoxicity and Proliferation of Human Fibroblasts

To evaluate fibroblast cytotoxicity and proliferation, we applied three complementary methodologies: (i) live/dead staining with Calcein AM and EthD-1, (ii) metabolic activity assessment using PrestoBlue, and (iii) CFSE-based cell division tracking by flow cytometry.

Fibroblasts were cultured in 24-well plates (1 × 10^3^ cells/well) with two different concentrations; one of 1.25 and another of 6.25 mg/mL of both stimulated (S1.25 and S6.25 mg/mL) and unstimulated (1.25 and 6.25 mg/mL) WJ-MSC secretome and exogenous EGF/FGF (10/40 ng/mL). The 1.25 and 6.25 mg/mL concentrations of secretome were chosen because they were the minimum doses at which fibroblasts proliferate and do not die, respectively. All conditions were prepared in plain medium DMEM, which was replaced every third day. The controls used plain medium with GFs or plain medium with 10% FSB. The evaluation of this test was at 24 and 72 h. Cells were washed twice with sterile HBSS at 4 °C for 10 min and then stained with 0.1 µL of 4 mM Calcein AM and 1 µL of 2 mM ethidium homodimer-1 (EthD-1) in 500 µL HBSS for 30 min at 37 °C. Calcein AM stains live cells green by intracellular esterase activity, while EthD-1 labels dead cells with red fluorescence. Cells were imaged using a Nikon Eclipse I30 microscope (4×, 10×, 20× objectives) (Nikon Corporation, Tokyo, Japan) and analyzed with NIS Elements software.

For metabolic activity assessment, fibroblasts were seeded in 96-well plates (1 × 10^4^ cells/well) under the same treatment conditions and maintained for 24, 72, and 168 h at 37 °C with 5% CO_2_. After incubation, cells were washed with HBSS, and 10 µL of PrestoBlue reagent mixed with 90 µL of Hank’s solution was added. Plates were incubated for 1 h at 37 °C with 5% CO_2_, and the absorbance intensity was measured at a wavelength of 570 nm using a Thermo Labsystems 354 Multiskan Ascent Microplate Reader (Thermo Fisher Scientific, Waltham, MA, USA). Readings were obtained using Ascent software version 2.6 for multiple layers.

On the other hand, fibroblast proliferation was also assessed using Carboxyfluorescein Succinimidyl Ester (CFSE) (BioLegend, San Diego, CA, USA). 1 × 10^6^ cells were labeled with 10 nM CFSE for 30 min. Cells were reseeding at 2 × 10^5^ cells per T25 flask under the same conditions (S1.25 and S6.25 mg/mL; 1.25 and 6.25 mg/mL; finally, GFs (EGF/FGF)

Cells were incubated at 37 °C with 5% CO_2_ for 5 days. At the end of the incubation, cells were harvested with trypsin–EDTA, resuspended in staining buffer, and analyzed by flow cytometry. The proliferation index was calculated using FlowJo software version 8.8.7 (BD, Franklin Lakes, NJ, USA).

### 2.7. In Vitro Wound-Healing Assay

Human fibroblasts were seeded in 6-well plates at a density of 5 × 10^5^ cells/well and allowed to grow for 48 h at 37 °C and 5% CO_2_ in DMEM supplemented with 10% FBS and 1% antibiotic (penicillin, streptomycin and amphotericin B, Gibco, Waltham, MA, USA). A ‘wound’ was created in the confluent cell monolayer using a sterilized pipette tip (1000 µL) and the dislodged cells removed by washing twice with PBS, before adding the media with different treatment solutions at various concentrations (1.25 and 6.25 mg/mL of unstimulated and S1.25 and S6.25 mg/mL growth factor–stimulated secretome of WJ-MSC) and exogenous EGF 10 ng/mL/FGF 40 ng/mL (GFs). All conditions were prepared in plain medium DMEM containing 1% antibiotic (penicillin, streptomycin, and amphotericin B, Gibco, Waltham, MA, USA), and the medium was replaced every third day. The control used medium plain, medium with GFs, and medium with FSB. Photomicrographs were taken with a Nikon Eclipse Ti-S attached with Nikon DS-Ri2 camera and NIS Elements v5.01 software (Nikon Corporation, Tokyo, Japan) at 0 and 4 days, and processed by ImageJ 1.54 software to determine the wound area and the rate of wound closure, calculated as a percent of the baseline wound area measured at 0 h. Experiments were performed in triplicate.

### 2.8. Effect of the WJ-MSC Secretome on Procollagen I Production and α-Smooth Muscle Actin Expression in Human Fibroblasts

Human fibroblasts (2 × 10^5^) were cultured in T25 flasks with DMEM supplemented with 10% FBS and 1% antibiotic (penicillin, streptomycin and amphotericin B, Gibco, Waltham, MA, USA) and exposed either to secretome from stimulated WJ-MSCs (1.25 or 6.25 mg/mL), secretome from unstimulated WJ-MSCs (S1.25 or S6.25 mg/mL), or to exogenous GFs (EGF and FGF). After 7 days at 37 °C and 5% CO_2_, cells were harvested with trypsin–EDTA, incubated with FITC-anti-procollagen I and PE-anti-α-smooth muscle actin antibodies (1:300) (BioLegend, San Diego, CA, USA), washed, fixed, and analyzed by flow cytometry (ATTUNE, BD, Franklin Lakes, NJ, USA). Data were processed using FlowJo v8.8.7 (BD, Franklin Lakes, NJ, USA).

### 2.9. Statistical Analysis

Data are presented as mean ± standard deviation (SD) from at least three independent experiments, measured as triplicates, and statistical analysis was performed with the GraphPad Prism 6 software (GraphPad, San Diego, CA, USA). An ANOVA with Bonferroni’s multiple comparison test comparison of multiple groups was done (*, *p* < 0.05, **, *p* < 0.01, ***, *p* < 0.001, ns, not significant).

## 3. Results

### 3.1. Isolation and Characterization of WJ-MSCs

WJ-MSCs were obtained using the explant method (Figure 1a1–a3). Non-adherent cells were discarded, while adherent cells proliferated to form a monolayer. As shown in Figure 1a4–a6, the cells exhibited an elongated morphology (indicated by arrows). Moreover, WJ-MSCs stimulated with growth factors (EGF and FGF) maintain their characteristic phenotype, expressing CD73, CD90, CD105, and CD44, while expressing low levels of HLA-ABC and lacking the hematopoietic markers CD34 and CD45, as well as HLA-DR and CD19 (Figure 1b).

The expression of fourteen growth factors (GFs) secreted into the culture medium by stimulated (W EGF/FGF) and unstimulated (WO EGF/FGF) WJ-MSCs was quantified using LEGENDplex analysis. As shown in Figure 2a, the concentration of total GFs in the W EGF/FGF secretome reached 5995 pg/mL, compared with 4278 pg/mL in the WO EGF/FGF secretome (*p* < 0.001), representing a 40.1% increase. Each of the 14 GFs was further analyzed and grouped according to their biological roles: maintenance of stemness (SCF, M-CSF, G-CSF, GM-CSF), proliferation of differentiated cells (HGF, FGF, EGF, PDGF-AA), angiogenesis (Ang-2, EPO, VEGF), and extracellular matrix deposition (PDGF-BB, TGF-α, TGF-β1). Significant differences were observed for G-CSF, HGF, VEGF, and TGF-α, which were elevated in the W EGF/FGF group compared with the WO EGF/FGF group (*p* < 0.05). The remaining GFs showed slightly higher concentrations in the stimulated group than in the unstimulated group, except for TGF-β1, which decreased upon stimulation. However, these differences were not statistically significant.

### 3.2. Secretome Preserves Fibroblast Viability and Enhances Proliferation

The cytotoxic effects of different secretome concentrations (S1.25, S6.25, 1.25, and 6.25 mg/mL) on human fibroblasts were assessed using calcein and ethidium homodimer staining, complemented by the PrestoBlue assay. As shown in Figure 3a, fibroblasts displayed the characteristic spindle-shaped morphology. Calcein-labeled cells (green) at 24 and 72 h showed increased density, with no dead cells detected (negative ethidium homodimer staining—red).

Metabolic activity measured at 24, 72, and 168 h correlated with observed cell density (Figure 3b). At 24 h, S1.25 did not differ significantly from FBS or GFs controls, whereas 1.25, 6.25, and S6.25 were significantly lower (*p* < 0.03 for 1.25; *p* < 0.008 for 6.25 and S6.25). At 72 h, S1.25 remained comparable to FBS but differed from GFs (*p* < 0.04) and from 1.25, 6.25, and S6.25 (*p* < 0.03). After 168 h, FBS exhibited the highest cell density, significantly exceeding all groups (*p* < 0.01 for GFs and S1.25; *p* < 0.001 for 1.25 and S6.25), while S1.25 matched GFs and surpassed 1.25, 6.25, and S6.25 (*p* < 0.04 and *p* < 0.03, respectively), indicating similar behavior to the controls at 24 and 72 h.

Finally, the effect of secretome concentration on fibroblast proliferation was evaluated with CFSE by flow cytometry (Figure 3c). At 24 h, all groups exhibited high mean fluorescence intensity (MFI), which decreased by 72 h. S1.25 did not differ from GFs or S6.25, indicating comparable proliferation induction, but showed reduced MFI relative to 1.25 and 6.25, reflecting increased cell divisions (*p* < 0.010). FBS differed significantly from S1.25 (*p* < 0.04)

### 3.3. WJ-MSC Secretome Enhances Fibroblast In Vitro Wound Healing

A wound closure in vitro assay was performed on a confluent human fibroblast monolayer to evaluate the effect of the secretome. Treatment with S1.25 markedly enhanced wound closure, as determined by measuring the cell-occupied area at the wound site on day 0 and after 4 days (Figure 4a). The percentage of cell invasion is shown in Figure 4b. The control group containing medium plain exhibited 37% cell occupancy with significant differences with the other groups (*p* < 0.001). S1.25 did not differ significantly from the FBS and GFs controls, showing occupancy rates of 90%, 89.9%, and 82%, respectively. Comparisons with the other groups (1.25, 6.25, and S6.25) revealed significant differences, indicating that S1.25 achieved approximately 90% wound closure after 4 days (*p* < 0.01).

### 3.4. WJ-MSC Secretome Promotes Type I Procollagen Expression

The effect of the WJ-MSC secretome on type I procollagen expression in human fibroblasts was assessed using immunohistochemistry (IHC) and flow cytometry. Fibroblasts were treated with different secretome concentrations (1.25, 6.25, S1.25, and S6.25 mg/mL) for 7 days. As shown in the photomicrographs (Figure 5a), all groups exhibited positive staining (brown). Flow cytometry revealed that S6.25 induced significantly higher collagen production compared with both the control and the other experimental groups (*p* < 0.01), while no significant differences were observed among the remaining groups (Figure 5b,c).

### 3.5. Secretome-Mediated Upregulation of Alpha-Actin in Human Fibroblasts

IHC photomicrographs revealed strong alpha-actin expression across all groups (Figure 6a). Flow cytometry analysis indicated that S1.25 and 6.25 did not differ significantly from each other but were considerably higher than 1.25, S6.25, and GFs (*p* < 0.01). The lowest alpha-actin mean fluorescence intensity (MFI) was observed in the FBS group, which differed highly significantly from all other groups (*p* < 0.001) (Figure 6b,c).

## 4. Discussion

The secretome is a mixture of soluble factors, proteins, extracellular vesicles, and micro RNAs released by mesenchymal or other cells. It is considered a more advantageous alternative to the direct use of cells in regenerative therapies, because this is safer, more standardisable, and easier to apply, while cells offer a dynamic and adaptable effect [34]. However, current evidence suggests that the regenerative benefits are primarily due to what they secrete, reason why we stimulated WJ-MSC with FGF and EGF to sustain proliferation, enhances fibroblast in vitro wound healing, and promote secretion of regenerative and immunomodulatory factors.

This study demonstrates that EGF/FGF-stimulated WJ-MSCs secrete a bioactive secretome that enhances fibroblast proliferation and wound closure at low concentrations (S1.25 mg/mL), highlighting its potential for regenerative applications. The umbilical cord is a valuable source of WJ-MSCs. Processing begins with the removal of blood vessels and fragmentation of connective tissue, which can then be isolated by enzymatic digestion or explant culture. The method of isolation has been evaluated by Dominici et al. 2006 [5] and Buyl et al. 2015 [35] by identifying the culture doubling time. However, comparisons between methods can be biased by factors such as donor variability, differences in protocols, and the composition of the culture medium. The methodology used for this study monitored homogeneity, in vitro formation of a cell monolayer, and morphological and phenotypic characteristics of the WJ-MSC culture. Furthermore, explant culture was found to be a reliable and effective method for isolating WJ-MSCs [36,37,38].

The value of WJ-MSCs lies not only in their reliable isolation but also in the stability of their phenotype under stimulation. We found that exposure of WJ-MSCs to EGF and FGF did not alter their mesenchymal profile. Flow cytometry at passages 1 and 2 confirmed positive expression of CD73, CD90, CD105, CD44, and low HLA-ABC, and negative expression of CD34, CD45, HLA-DR, and CD19. These results are consistent with observations in skin-derived MSCs stimulated with EGF and FGF, which retained CD90, CD105, and CD73 expression while lacking CD45, CD14, and HLA-DR [36]. In our study, this phenotype was preserved in 95% of cells across passages 1 and 2. However, from passage 3 onwards, a small CD34-positive population emerged, suggesting that growth factor stimulation may progressively influence the phenotypic profile of MSCs [36,39].

Given the stability of the WJ-MSC phenotype, we next evaluated the functional impact of growth factor stimulation. Salehinejad et al. 2013 [40] showed that culturing umbilical cord blood MSCs with EGF, and to a lesser extent FGF, enhances proliferation and expansion, consistent with our findings. We observed that the secretome of WJ-MSCs stimulated with EGF and FGF exhibited a 40.1% increase in G-CSF, HGF, VEGF, and TGFα compared to the unstimulated secretome. Correspondingly, fibroblast proliferation and matrix production were greater when exposed to the stimulated secretome. This observation is particularly relevant as the regenerative potential of MSCs is primarily mediated by their paracrine activity, which encompasses a broad repertoire of bioactive molecules, including proteins, peptides, lipids, extracellular vesicles, RNA, cytokines, chemokines, immunomodulatory molecules, and growth factors [37,41]. Notably, the MSC-derived secretome can be concentrated and lyophilized without losing functionality, offering a practical alternative to cell-based therapies that require continuous culture and cryopreservation. Moreover, secretome can be delivered via direct injection, incorporated into scaffolds for controlled release, or employed to direct differentiation of resident cells toward specific phenotypes in damaged tissues [42,43]. Considering its potential applications, we assessed how different concentrations of stimulated secretome influence fibroblast proliferation, morphology, and metabolic activity. The S1.25 concentration preserved both proliferation and fibroblastoid morphology. After seven days, although the FBS control exhibited the highest cell number—likely due to its optimization for fibroblasts—S1.25 induced proliferation comparable to the GFs control and higher than the 1.25, 6.25, and S6.25 groups. These results suggest that lower secretome doses promote proliferation, whereas higher concentrations (6.25) may exert inhibitory or saturating effects, indicating a dose-dependent response. While further studies were warranted, our findings suggest that low concentrations of stimulated secretome (S1.25) enhance proliferation, whereas high doses may trigger negative feedback or early senescence in vitro [44].

Considering the dose-dependent effects observed, we next examined the contribution of individual growth factors within the secretome to its regenerative potential. G-CSF plays a key role in cell proliferation, survival, and migration, and acts as an enhancer of hematopoietic and immunomodulatory processes [45]. HGF is critical for epithelial tissue regeneration and protection against apoptosis [46,47], while VEGF is a central regulator of angiogenesis, essential for graft integration and healing. TGFα supports cell proliferation and repair, complementing the functions of the other factors [48]. Tissue regeneration, however, relies not on a single factor but on the synergistic interaction of multiple bioactive molecules. The simultaneous presence of G-CSF, HGF, VEGF, and TGFα at significant concentrations suggests a secretory profile with particularly favorable regenerative potential. These findings align with previous reports indicating that the secretome is more effective than isolated factors, as it more faithfully reproduces the complex microenvironment required for tissue repair [26,37,49]. Notably, TGF-β1 is associated with fibrosis and excessive extracellular matrix deposition, which can limit tissue regeneration through fibrotic scarring [50]. In our study, stimulated secretome TGF-β1 levels were reduced; although this did not reach statistical significance, it suggests a more pro-regenerative environment with lower fibrogenic potential. Overall, a secretome enriched in angiogenic, proliferative, and stem cell–related factors could enhance applications in bone and cartilage regeneration, wound healing, and cardiovascular therapies, particularly due to the presence of VEGF and HGF.

On the other hand, the expression of procollagen and alpha-actin was selected as an indicator of fibroblast function, since procollagen reflects extracellular matrix synthesis activity typical of fibroblasts. In contrast, alpha-actin is associated with cytoskeletal organization and contractile capacity, characteristic features of activated fibroblasts or myofibroblasts during tissue remodeling [51]. The fibroblasts stimulated with secretome S1.25 did not show an increase in procollagen expression, but only S6.25. However, alpha-actin expression was elevated at the S1.25 and 6.25 concentrations but not in S6.25; this could indicate that the cells are not transforming into myofibroblasts, according to what was reported by Tutuianu R et al. 2021, which demonstrates that the secretome does not induce differentiation into myofibroblasts [52]. In addition, Ronen Schuster et al. 2023 defined myofibroblasts as large cells with short cellular extensions, which differ in shape from fibroblasts [53]. Therefore, myofibroblasts must express alpha-actin and be producers of procollagen. It should also be noted that the shape of the fibroblasts did not vary across treatments [52,53,54].

Finally, we investigated how growth factor–stimulated secretomes influence fibroblast wound closure. The EGF/FGF-stimulated secretome enabled cells to occupy approximately 90% of the wound area in vitro within four days. Comparable studies have been conducted using MSC-derived exosomes [52] or secretomes treated with interferon, the phorbol ester PMA, and lipopolysaccharide [54], which delayed wound closure. In vitro wound closure occurs through the migration and proliferation of cells into an artificially created open space, modulated by autocrine and paracrine signals. It is increased by the secretion of factors such as TGF-β, VEGF, FGF, and EGF, which modulate the speed of closure [55]. We found an increase in the concentration of VEGF in the secretome of stimulated samples, which could be the reason for the wound closure.

Another studies have demonstrated that the conditioned medium derived from WJ-MSCs promoted wound healing and closure in an in vivo murine model via paracrine mechanisms and this effect was caused to the bioactive molecules present in the conditioned medium, which stimulated fibroblast migration toward the injury site, also the secretome enhances the repair of radiation-induced wounds by increasing fibroblast viability in a rat model and accelerated acute wound healing by promoting fibroblast and keratinocyte proliferation, exerting pro-angiogenic and anti-inflammatory effects, and reducing oxidative stress [20,56,57].

The use of WJ-MSC secretome improves various aspects of dermal regeneration. In the present study, we propose that enriching the secretome by preconditioning Wharton’s jelly mesenchymal stem cells with EGF and FGF could further enhance its regenerative potential. These growth factors positively modulate the MSC secretory profile, increasing the release of bioactive molecules involved in proliferation, migration, angiogenesis, and extracellular matrix remodeling, processes critical for effective cutaneous repair [58]. Although our study demonstrates that the secretome derived from EGF- and FGF-preconditioned WJ-MSCs enhances fibroblast proliferation, migration, and functional markers relevant to dermal regeneration, several limitations should be considered: we did not directly compare the effects of applying EGF and FGF directly to fibroblasts versus using the secretome of growth factor-treated WJ-MSCs. Direct growth factor treatment can stimulate proliferation and migration. Still, it may not reproduce the complex paracrine signaling and synergistic interactions provided by the MSC secretome, which contains cytokines, chemokines, and extracellular vesicles. The composition and potency of the secretome can vary depending on donor variability, MSC passage number, and culture conditions. Finally, long-term effects on fibroblast phenotype, including potential differentiation into myofibroblasts or changes in extracellular matrix production, were not evaluated. Future studies directly comparing both approaches and standardizing secretome production are necessary to fully elucidate the optimal strategy for promoting dermal repair.

## 5. Conclusions

Taken together, these results highlight the potential of the S1.25 secretome as a safe and effective promoter of fibroblast proliferation, exhibiting behavior comparable to conventional growth factors. Our results suggest that secretome S1.25 could be a promising alternative or complement in cell culture, with potential applications in skin wound healing, connective tissue regeneration, and the development of bioactive biomaterials designed for controlled release of secretome.

## Figures and Tables

**Figure 1 biomedicines-13-02863-f001:**
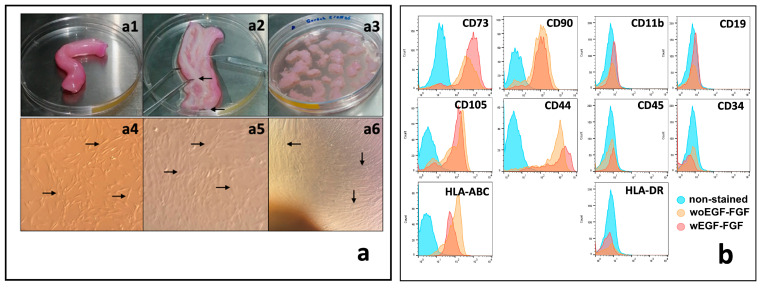
Isolation and characterization of human WJ-MSCs. (**a**) Processing of the umbilical cord and culture of WJ-MSCs. (**a1**) 10 cm length umbilical cord specimens were collected. (**a2**) Blood vessels (arrows) were identified and dissected from the umbilical cord specimens. (**a3**) After blood vessels removal, Wharton jelly was fragmented into 2 × 2 mm pieces and cultured by explant. (**a4**) In short-term cultures, MSCs showed great adherence to the culture plate with a fusiform appearance (arrows). (**a5**) In the medium-term cultures, MSCs showed a typical change in their morphology, becoming wider and with cytoplasmic projections (arrows). (**a6**) In the long term, MSCs reached confluence greater than 80%, forming a monolayer of compacted cells (arrows). (**b**) Flow cytometry phenotype analysis of WJ-MSCs. Histograms show the expression of typical positive and negative markers of MSCS in cultured cells at passage 2, stimulated or unstimulated with EGF and FGF (ns). Magnification of (**a4**–**a6**) = 40×.

**Figure 2 biomedicines-13-02863-f002:**
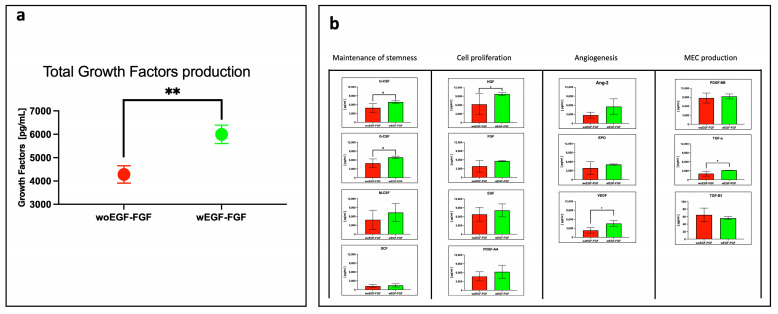
Determination of some growth factors produced by non-stimulated and stimulated WJ-MSCs. Growth factor quantification was obtained using the LEGENDplex Human growth factor multiplex assay (BioLegend, USA). (**a**) Comparison of total growth factors production by WJ-MSCs. WJ-MSCs cultured in the presence of EGF-FGF stimulation showed a significantly greater total production of growth factors than unstimulated ones (** *p* < 0.01). (**b**) Comparison of each of the 14 Growth Factors produced by WJ-MSCs. Growth Factors were grouped by biological role, and significant differences were observed for G-CSF, HGF, VEGF, and TGF-α (* *p* < 0.05). n = 3 independent experiments, each performed in triplicate.

**Figure 3 biomedicines-13-02863-f003:**
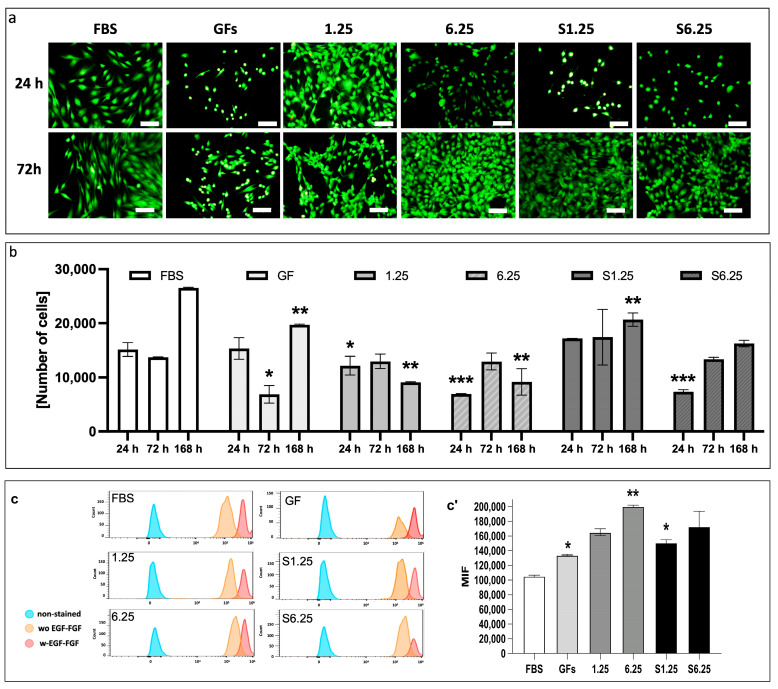
Analysis of cytotoxicity, metabolic activity, and proliferation in human dermal fibroblasts cultured with the secretome of WJ-MSCs. (**a**) Live and dead assay. Calcein green fluorescent stain shows live cells, and ethidium homodimer red fluorescent stain denotes dead cells. (**b**) Prestoblue colorimetric assay. Metabolic activity was measured at 24, 72, and 168 h. At 24 h, S1.25 did not differ significantly from FBS or GFs controls, whereas 1.25, 6.25, and S6.25 were significantly lower (* *p* < 0.03, ** *p* < 0.01, and *** *p* < 0.008). At 72 h, S1.25 remained comparable to FBS but differed from GFs (*p* < 0.04) and from 1.25, 6.25, and S6.25 (*p* < 0.03). After 168 h, FBS showed the highest cell density, significantly exceeding that of all groups. (**c**) Histograms of CFSE in human fibroblasts with different concentrations of WJ-MSC secretome obtained from cells stimulated and unstimulated with FGF-EGF. (**c’**) MIF graphic of CFSE. The graphic shows that FBS has lower MIF levels, indicating more cell divisions. GFs and S1.25 exhibited similar patterns in their MIF. * *p* < 0.04 and ** *p* < 0.01). n = 3 independent experiments, each performed in triplicate. Bars = 50 μm.

**Figure 4 biomedicines-13-02863-f004:**
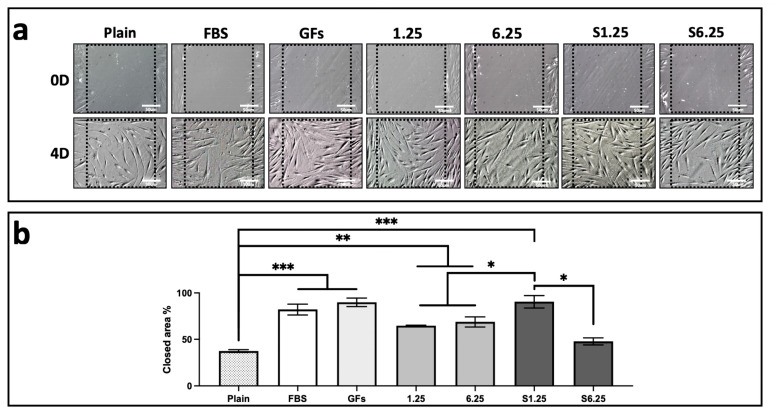
Scratch wound assay on fibroblast cultured with WJ-MSC secretome. (**a**) DIC photomicrographs of fibroblast cultures at day 0 and day 4 after scratching. (**b**) Quantification of scratch area occupied by fibroblasts expressed as a percentage, (*** *p* < 0.001, ** *p* < 0.01, * *p* < 0.05). n = 3 independent experiments, each performed in triplicate. Bars = 50 μm.

**Figure 5 biomedicines-13-02863-f005:**
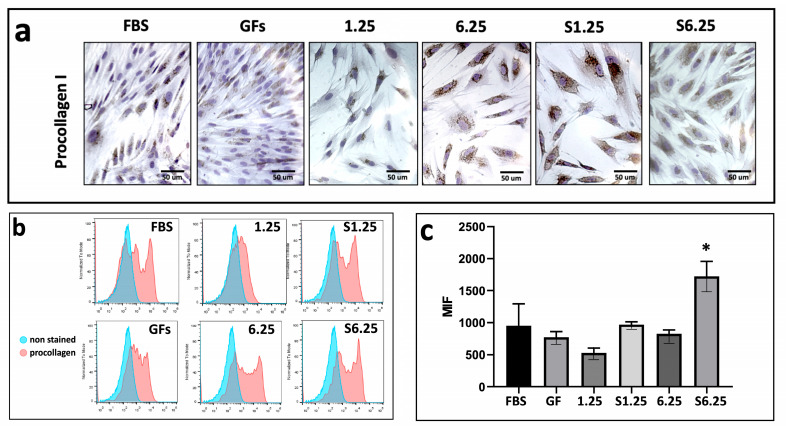
Procollagen expression on fibroblasts cultured with WJ-MSC secretome. (**a**) Immunohistochemistry demonstration of procollagen-I after 7 days of culture. (**b**) Histograms of Procollagen marked in human fibroblasts with different concentrations of WJ-MSC secretome. (**c**) Graphic of MIF procollagen I expression after 7 days of culture. Flow cytometry revealed that S6.25 showed a significantly expression of procollagen than other experimental groups (* = *p* < 0.01). n = 3 independent experiments, each performed in triplicate. Bars = 50 μm.

**Figure 6 biomedicines-13-02863-f006:**
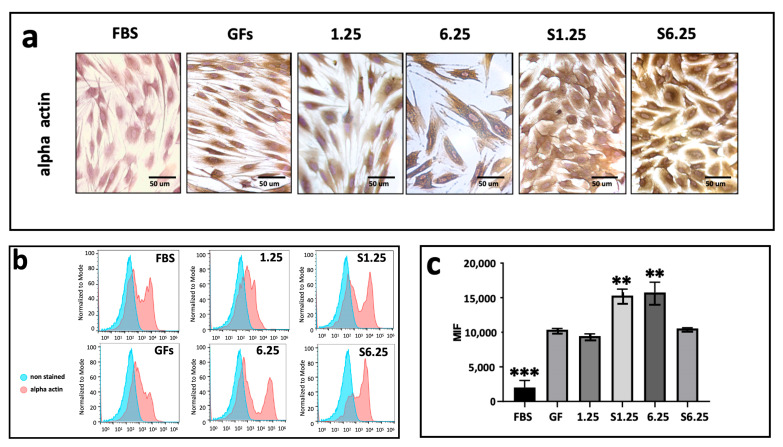
Alpha actin expression on fibroblasts cultured with WJ-MSC secretome. (**a**) Immunohistochemistry demonstration of alpha actin after 7 days of culture. (**b**) Histograms of alpha actin marked in human fibroblasts with different concentrations of WJ-MSC secretome. (**c**) Graphic of MIF alpha actin expression after 7 days of culture. The lowest alpha-actin expression was present in FBS group (*** *p* < 0.001) whereas S1.25 and 6.25 groups showed a higher expression (** = *p* < 0.01). n = 3 independent experiments, each performed in triplicate.

## Data Availability

Data will be made available upon request to the authors.

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
