# Peer review of "Growth Factor-Primed WJ-MSC Secretome Enhances Fibroblast Expansion In Vitro"

_biomedicines, 2025, doi:10.3390/biomedicines13122863_

Round 1
Reviewer 1 Report
Comments and Suggestions for Authors
The authors of the manuscript titled "Growth Factor Primed WJ-MSC Secretome Enhances Fibroblast Expansion in Vitro" investigated the effect of WJ-MSC-derived secretome on primary fibroblasts. They treated WJ-MSC with EGF and FGF, collected the supernatant, lyophilized it, and discovered that the amount of various growth factors was significantly higher in the treated group than in the untreated WJ-MSC group. The prepared mixtures were added to primary fibroblasts at two concentrations. To demonstrate the effect on viability, live-dead staining was performed. Low concentrations of secretome (treated WJ-MSC) had positive effects on cell growth. A similar development was observed in a scratch test simulating wound healing. Fibroblasts treated with the secretome of WJ-MSCs showed increased expression of procollagen 1, while alpha-actin was upregulated in fibroblasts treated with the low concentration. In general, the quality of the figures (e.g., image orientation, figure description, labeling of multiple panels, etc.) is poor. Numerous typographical errors (e.g., mL vs. ml, etc.) and spelling errors were found throughout the manuscript. The manuscript appears to have been poorly written and lacks attention to editing and proofreading. The research area is highly relevant, and the manuscript offers interesting insights into future treatment strategies.
The following shall be considered be fore publication:
- Please check the spaces between the unit and the number (e.g. abstract line 19) in the whole manuscript
- Please check for consistent spelling of ml or mL (e.g. methods part line 19 and 48) in the whole manuscript
- It would be nice to introduce abbreviations in a more consistent way: e.g. FGF is abbreviated but the bFGF has to be searched in the abbreviations list; Introduction of bFGF as basic Fibroblast Growth
- It would be nice to get the connection in the introduction part of Wharton´s jelly mesenchymal stem cells are expressing typical mesenchymal stem cell marker genes.
- Were the EGF and FGF dissolved in medium or in another solvent? If so, was this solvent used in the control group?
- It is not provided which antibiotics were used. Only the term “antibiotics” is not sufficient.
- What was the amount of supernatant that was used to produce the lyophilized supernatant and what was the yield to generate a final concentration of 100 mg/ml?
- Please add final concentration of Calcein and Ethidiumhomodimer: 1 µl is not a concentration
- It would be good to provide more information about the antibodies like the Cat. No.
- Please provide some information about the donors for the fibroblast isolation (age, gender distribution, smoking)
- Different spelling type of 106, 10^5, 105.
- Why were 10 ng/ml and 40 ng/ml EGF/FGF used? It should be explained were this information come from.
- Was there a medium change during the growth time of the viability assays and scratch assays, or had the cells over 4 days the same medium?
- Figure 1: Labeling of the histogram axis is missing; No labeling in a, of the brightfield images.
- Figure 2: There is no labeling of the multi-panel; In the text it is written of Fig 2a and b but the figure does not contain the labeling; Moreover, it would be beneficial when the figure description would contain number of experiments, technical replicates and so on.
- It is not getting clear why the concentration of 1.25 mg/ml and 6.25 mg/ml come from. It would be beneficial to explain why these concentrations were used.
- Figure 3 and figure 4 are in the same chapter, is it necessary to have them in different figures. It would be good to combine them.
- In figure 4 the multi-panel labeling is missing. Moreover, no labeling of the histograms.
- Figure 5: The microscopy images are not aligned. No multi-panel labeling; Here a plane control was added, why this plane control was not included in the cytotoxicity chapter? In the FBS condition; Medium without any additions called plain.
- Sometimes in the graph labeling/x-axis there is a space between the conditions sometimes not (image: S 1.25; graph: S1.25). Please change to a consistent way.
- Figure 6: No numbering in a, b, c, etc., It is announced that there are significant differences between the S6.25 and the other experimental groups but it is not shown in the graph.
- It would be good to explain why procollagen and alpha actin are typical marker to look at in fibroblast functions.
- It is written that the expression of type 1 collagen was analyzed. But in the methods part and in the figure, Procollagen 1 was analyzed. A more specific writing would be good. The statistics in the measured graph is missing. Why was the day 7 time point chosen to analyze the Procollagen expression?
- In Figure 7 the same points as before. No multi-panel labeling, no statistics in the quantification, no information about replicates.
- For all of the figures/graphs should be considered, that error bars should always be shown in both directions.
- In the discussion part the writing style of et al. is inconsistent. Sometimes it is written in italic sometimes not.
- In the discussion part is a writing mistake when alpha actin is compared. Alpha actin was increase based on the results in 6.25 and S1.25 and not in S6.25.
- A comparison to other studies that analyzed the secretome effect on other cells would fit in the discussion part.
- It would be good to add a limitations part to the discussion. A part about the comparison of the direct treatment of growth factors vs the secretome of growth factor treated WJ-MSC would increase the quality of the discussion part.
- The abbreviation list has to be revised. Please use a consistent format of the table. They are not in line. Moreover; the list is usually in alphabetical order.
Author Response
Reviewer 1
- Please check the spaces between the unit and the number (e.g. abstract line 19) in the whole manuscript.
- We appreciate your comment, and we have already done it throughout the manuscript.
- Please check for consistent spelling of ml or mL (e.g. methods part line 19 and 48) in the whole manuscript
- Again, we appreciate your suggestions, and we correct it in the manuscript.
- It would be nice to introduce abbreviations in a more consistent way: e.g. FGF is abbreviated but the bFGF has to be searched in the abbreviations list; Introduction of bFGF as basic Fibroblast Growth
- Many thanks for your observation. We corrected the abbreviations in the manuscript.
- It would be nice to get the connection in the introduction part of Wharton´s jelly mesenchymal stem cells are expressing typical mesenchymal stem cell marker genes.
- We have added more detailed to get the connection in the introduction part of Wharton´s jelly mesenchymal stem cells are expressing typical mesenchymal stem cell marker genes in the Introduction section (page 2, lines 7-12).
- Were the EGF and FGF dissolved in medium or in another solvent? If so, was this solvent used in the control group?
- Many thanks. We've added that the EGF and FGF were dissolved in plain medium (DMEM-F12) and in the control group were DMEM-F12/FSB and DMEM-F12/GFs in the Methods section (page 3, lines 26-27).
- It is not provided which antibiotics were used. Only the term “antibiotics” is not sufficient.
- We have revised the entire manuscript to include the corresponding information about of antibiotic (penicillin, streptomycin and amphotericin B, Gibco, USA), in the Methods section (page 3, lines 11, 15 and 18).
- What was the amount of supernatant that was used to produce the lyophilized supernatant and what was the yield to generate a final concentration of 100 mg/ml?
- We've added in the text that 0.3235 g of lyophilized powder were obtained of 10 mL of supernatant of WJ-MSC. The concentration of 100 mg/mL is the minimal concentration to be detected for the LEGENDplex, (page 3, lines 28-29; lines 47-48).
- Please add final concentration of Calcein and Ethidium homodimer: 1 µl is not a concentration
- We have revised the manuscript to include the corresponding final concentration of 4 mM Calcein AM and 1 µL of 2 mM ethidium homodimer-1 (page 4, lines 33-34).
- It would be good to provide more information about the antibodies like the Cat. No.
- We agree with the suggestion. We've added the catalog numbers in the text (page 3, lines 36-40).
- Please provide some information about the donors for the fibroblast isolation (age, gender distribution, smoking)
- We appreciate your comment. We´ve added all tissues were voluntarily donated by healthy male with an age range between 25 and 40 years old, in the Methods section (page 4, lines 7-8)
- Different spelling type of 106, 10^5, 105.
- We have revised the entire manuscript to check different spelling type of 106, 10^5, 105
- Why were 10 ng/ml and 40 ng/ml EGF/FGF used? It should be explained were this information come from.
- We appreciate your observation and we´ve added in Methods section (page 3, lines 25-26) “according to Park et al (2009) these concentrations were effective in inducing stem cell proliferation”.
- Was there a medium change during the growth time of the viability assays and scratch assays, or had the cells over 4 days the same medium?
- We are gratefully for your question. We've added, “all conditions were prepared in plain medium DMEM and it was replaced every third day. The control used medium with GFs and medium with 10% FSB,” in the Methods section (page 4 lines 30-32; page 5 lines 14-15)
- Figure 1: Labeling of the histogram axis is missing; No labeling in a, of the brightfield images.
- We corrected the figure and the caption in the text. (page 6 lines 1-12)
- Figure 2: There is no labeling of the multi-panel; In the text it is written of Fig 2a and b but the figure does not contain the labeling; Moreover, it would be beneficial when the figure description would contain number of experiments, technical replicates and so on.
- We’ve added n = 3 independent experiments, each performed in triplicate in all the captions
- It is not getting clear why the concentration of 1.25 mg/ml and 6.25 mg/ml come from. It would be beneficial to explain why these concentrations were used.
- We appreciate your feedback. In a pilot study, we evaluated different secretome concentrations in fibroblast proliferation and observed that fibroblasts died at a concentration of 12.5 mg/mL, as analyzed using the PrestoBlue technique. Therefore, we developed a dose/response curve, where we found that the minimum dose at which fibroblasts proliferated was 1.25 mg/mL. Furthermore, we observed that the minimum dose at which cells did not die, it was 6.25 mg/mL. Therefore, we used these doses to perform the experiments in this work. These results are shown in the attached table. In the text of the article, we have included a brief explanation in page 4 lines 28-30.
- Figure 3 and figure 4 are in the same chapter, is it necessary to have them in different figures. It would be good to combine them.
- We agree with your suggestion, and we combined Figure 3 and figure 4. Now there is only one figure (figure 3).
- In figure 4 the multi-panel labeling is missing. Moreover, no labeling of the histograms.
- We combined this figure with the figure 3
- Figure 5: The microscopy images are not aligned. No multi-panel labeling; Here a plane control was added, why this plane control was not included in the cytotoxicity chapter? In the FBS condition; Medium without any additions called plain.
- We corrected the figure and the caption in the text. Now it is figure 4. (page 9 lines 4-8). By the other hand, we do not consider this control in cytotoxicity experiments because freshly harvested cells are immediately placed in its corresponding medium. If we placed freshly harvested cells in plain medium, the percentage of cell adhesion to the plastic would be minimal due to the lack of nutrients, resulting in a significant amount of cell death, so it would not be a good control. Furthermore, the cells in the control “plain medium” in Figure 5, that now is Figure 4, were cultured with medium supplemented with 10% FBS for 48 hours prior to wounding, and after that, plain medium was subsequently added to the culture as a negative control for proliferation.
- Sometimes in the graph labeling/x-axis there is a space between the conditions sometimes not (image: S 1.25; graph: S1.25). Please change to a consistent way.
- We corrected all the figures and the captions in the text.
- Figure 6: No numbering in a, b, c, etc., It is announced that there are significant differences between the S6.25 and the other experimental groups but it is not shown in the graph.
- Figure 6 is now figure 5. We correct the figure and the caption, page 10, lines 1-9.
- It would be good to explain why procollagen and alpha actin are typical marker to look at in fibroblast functions.
- We appreciate your question and we´ve added in Discussion section (page 13, lines 35-39) “the expression of procollagen and alpha-actin was selected as an indicator of fibroblast function, since procollagen reflects extracellular matrix synthesis activity typical of fibroblasts. In contrast, alpha-actin is associated with cytoskeletal organization and contractile capacity, characteristic features of activated fibroblasts or myofibroblasts during tissue remodeling (53).
- It is written that the expression of type 1 collagen was analyzed. But in the methods part and in the figure, Procollagen 1 was analyzed. A more specific writing would be good. The statistics in the measured graph is missing. Why was the day 7 time point chosen to analyze the Procollagen expression?
- Type I procollagen was assessed at 7 days of culture, as this period represents an early stage of extracellular matrix synthesis by mesenchymal cells and fibroblasts. Previous studies have shown that COL1A1 expression and type I procollagen secretion significantly increase between days 7 and 11 of culture (Mäkelä JK, Vuorio T, Vuorio E. Growth-dependent modulation of type I collagen production and mRNA levels in cultured human skin fibroblasts. Biochim Biophys Acta. 1990 Jun 21;1049(2):171-6. doi: 10.1016/0167-4781(90)90037-3. PMID: 2364107). We have already changed the expression of type 1 collagen by Procollagen 1, in Results section (page 9, line 10)
- In Figure 7 the same points as before. No multi-panel labeling, no statistics in the quantification, no information about replicates.
- Figure 7 is now figure 6. We correct the figure and the caption, page 10, line 17; page 11 lines 1-8.
- For all of the figures/graphs should be considered, that error bars should always be shown in both directions.
- We appreciate your suggestion and we have already done it.
- In the discussion part the writing style of et al. is inconsistent. Sometimes it is written in italic sometimes not.
- We have already done it throughout the manuscript.
- In the discussion part is a writing mistake when alpha actin is compared. Alpha actin was increase based on the results in 6.25 and S1.25 and not in S6.25.
- We corrected the text in the Discussion section (page 12, line 41)
- A comparison to other studies that analyzed the secretome effect on other cells would fit in the discussion part.
- We agree with the suggestion and we´ve added in Discussion section (page 13, lines 6-19) “Another studies have demonstrated that the conditioned medium derived from WJ-MSCs promoted wound healing and closure in an in vivo murine model via paracrine mechanisms and this effect was caused to the bioactive molecules present in the conditioned medium, which stimulated fibroblast migration toward the injury site, also the secretome enhances the repair of radiation-induced wounds by increasing fibroblast viability in a rat model and accelerated acute wound healing by promoting fibroblast and keratinocyte proliferation, exerting pro-angiogenic and anti-inflammatory effects, and reducing oxidative stress (58-60).
The use of WJ-MSC secretome improves various aspects of dermal regeneration. In the present study, we propose that enriching the secretome by preconditioning Wharton’s jelly mesenchymal stem cells with EGF and FGF could further enhance its regenerative potential. These growth factors positively modulate the MSC secretory profile, increasing the release of bioactive molecules involved in proliferation, migration, angiogenesis, and extracellular matrix remodeling, processes critical for effective cutaneous repair (61).”
- It would be good to add a limitations part to the discussion. A part about the comparison of the direct treatment of growth factors vs the secretome of growth factor treated WJ-MSC would increase the quality of the discussion part.
- We agree with the suggestion and we´ve added in Discussion section (page 13, lines 20-33) “Although our study demonstrates that the secretome derived from EGF- and FGF-preconditioned WJ-MSCs enhances fibroblast proliferation, migration, and functional markers relevant to dermal regeneration, several limitations should be considered: we did not directly compare the effects of applying EGF and FGF directly to fibroblasts versus using the secretome of growth factor-treated WJ-MSCs. Direct growth factor treatment can stimulate proliferation and migration. Still, it may not reproduce the complex paracrine signaling and synergistic interactions provided by the MSC secretome, which contains cytokines, chemokines, and extracellular vesicles. The composition and potency of the secretome can vary depending on donor variability, MSC passage number, and culture conditions. Finally, long-term effects on fibroblast phenotype, including potential differentiation into myofibroblasts or changes in extracellular matrix production, were not evaluated. Future studies directly comparing both approaches and standardizing secretome production are necessary to fully elucidate the optimal strategy for promoting dermal repair.”
- The abbreviation list has to be revised. Please use a consistent format of the table. They are not in line. Moreover, the list is usually in alphabetical order.
- We have revised the entire manuscript to correct the abbreviations and we done a table.

Reviewer 2 Report
Comments and Suggestions for Authors
1- Do the growth factors added to the cell culture medium not affect the measured value after stimulation? In other words, is the amount of growth factor secretion observed in Figure 2 not the same as the amount added to the culture medium?
2- Is the addition of the two growth factors mentioned to the medium stimulating the secretion of another growth factor, which can help resolve the bias in the previous comment?
3- What do you know about the reason for the decrease in survival after a concentration of 6.25 (Figure 4)
4- In your opinion, the findings of this study will be most effective in which of the stages of wound healing in living organisms? The following articles are helpful
https://link.springer.com/article/10.1007/s10924-025-03543-2
https://www.mdpi.com/2227-9059/11/9/2526
5- In some cases, there are spelling errors such as index, please correct. Page 4, second line.
6- It seems that the scratch test is introduced as wound healing in the study. The scratch test is more indicative of cell migration than wound healing. Please explain.
7- In Figure 7, what do you think is the reason for the non-significance of the difference between groups S1.25 and S6.25?
Author Response
Reviewer 2
1- Do the growth factors added to the cell culture medium not affect the measured value after stimulation? In other words, is the amount of growth factor secretion observed in Figure 2 not the same as the amount added to the culture medium?
- We appreciate your comment. The growth factors added to the cell culture medium does not affect the measured value after stimulation, because this stimulating medium was removed from the cells, and subsequently the cells were washed with HBSS solution and the supernatant is obtained in the plain medium (DMEM-F12). Therefore, the measurement value in figure 2 is on this medium plain without factors.
2- Is the addition of the two growth factors mentioned to the medium stimulating the secretion of another growth factor, which can help resolve the bias in the previous comment?
- Thank you for your interesting question. We only observed significant differences in the secretion of VEGF, GCSF, HGF, and TGFα, and it is worth noting that EGF and FGF did not have significant values in the secretome of stimulated WJ-MSCs. In this regard, the concentrations of EGF and FGF used to stimulate WJ-MSCs are 10 ng/mL and 40 ng/mL, respectively; however, the observed concentrations of EGF and FGF are in the order of picogram/mL. Thus, the concentrations of EGF and FGF added to the culture medium to stimulate the cells differ from those shown in Figure 2. Furthermore, as previously mentioned, the stimulation medium containing EGF and FGF is removed after 48 hours, and the cells are washed with Hanks' solution. The growth factor stimulation medium can indeed induce the secretion of other growth factors that we did not evaluate in our study, a situation that we will investigate in the future.
3- What do you know about the reason for the decrease in survival after a concentration of 6.25 (Figure 4)
- High concentrations of growth factors can induce undesirable effects in cultured cells, including cell death, oxidative stress, imbalance between ERK and JNK/p38, and senescence due to replicative stress (p53/p21). In our pilot study, high concentrations of the secretome (12.5 mg/mL) induced cell death. This may be due to the high concentrations of factors such as TNFα, TGFβ, EGF, FGF, VEGF, PDGF, and NGF. In our experiments, significant concentrations of VEGF were detected, which can cause cell death at 12.5 mg/mL. We evaluated only 14 factors; it is possible that other factors, such as TNF-α and TGF-β, may also be increased, leading to cell death at high secretome concentrations. At present, we are evaluating other growth factors and cytokines in supernatants from WJ-MSC in different experiments that will be published later.
- Hong X, Yu Z, Chen Z, Jiang H, Niu Y, Huang Z. High molecular weight fibroblast growth factor 2 induces apoptosis by interacting with complement component 1 Q subcomponent-binding protein in vitro. J Cell Biochem. 2018 Nov;119(11):8807-8817. doi: 10.1002/jcb.27131. Epub 2018 Aug 29. PMID: 30159917; PMCID: PMC6220755.
- Zhao X, Dai W, Zhu H, Zhang Y, Cao L, Ye Q, Lei P, Shen G. Epidermal growth factor (EGF) induces apoptosis in a transfected cell line expressing EGF receptor on its membrane. Cell Biol Int. 2006 Aug;30(8):653-8. doi: 10.1016/j.cellbi.2006.04.004. Epub 2006 Apr 27. PMID: 16750403.
4- In your opinion, the findings of this study will be most effective in which of the stages of wound healing in living organisms? The following articles are helpful
https://link.springer.com/article/10.1007/s10924-025-03543-2 https://www.mdpi.com/2227-9059/11/9/2526
- We appreciate the articles you provided. It has been demonstrated that the secretome of WJ-MSC modulates the immune response by inducing an anti-inflammatory environment. Also, secretome promotes tissue repair through angiogenic factors such as FGF, HGF, and VEGF, which stimulate fibroblast proliferation and the synthesis of extracellular matrix. Therefore, the secretome of WJ-MSC may diminished the inflammatory stage of wound healing. Also, it has been demonstrated that chronic wounds have an increment of the inflammatory stage. Therefore, the antinflammatory effect of the secretome could improve the wound healing by attenuate the inflammatory stage. We wrote a little paragraph in the discussion about this issue (page 13, lines 6-8).
https://www.mdpi.com/2227-9059/11/9/2526
5- In some cases, there are spelling errors such as index, please correct. Page 4, second line.
We appreciate your comment and we corrected this error.
6- It seems that the scratch test is introduced as wound healing in the study. The scratch test is more indicative of cell migration than wound healing. Please explain.
- In vitro wound tests approximate in vivo wounds. Obviously, in vitro wounds do not exhibit all the phases of in vivo wound closure.
If in vitro wound closure is assessed between 12 and 16 hours, it is most likely migration, as proliferation takes longer to induce wound closure. Closure observed after 48 hours or more may involve both processes, migration and proliferation.
It is also essential to consider that in in vivo wounds, cell migration occurs, primarily fibroblasts. The proliferation process is slightly later, but it also appears. It is evident from a large number of experiments that both processes are present.
- Kramer N, Walzl A, Unger C, Rosner M, Krupitza G, Hengstschläger M, Dolznig H. In vitro cell migration and invasion assays. Mutat Res. 2013 Jan-Mar;752(1):10-24. doi: 10.1016/j.mrrev.2012.08.001. Epub 2012 Aug 23. PMID: 22940039.

Round 2
Reviewer 1 Report
Comments and Suggestions for Authors
There are no further comments
Reviewer 2 Report
Comments and Suggestions for Authors
The manuscript is well-revised and can be accepted for publication.